# Distinct landscapes of deleterious variants in DNA damage repair system in ethnic human populations

Zixin Qin, Teng Huang, Maoni Guo, San Ming Wang

**Deleterious variants in DNA damage repair (DDR) system can cause genome instability and increase cancer risk. In this study, we analyzed the deleterious variants in DDR system in 16 ethnic human populations. From the genetic variants in 169 DDR genes involved in nine DDR pathways collected from 158,612 individuals of different ethnic background, we identified 1,781 deleterious variants in 81 DDR genes in eight DDR pathways (https://genemutation.fhs.um.edu.mo/dbddr-global/). Our analysis showed although the quantity of deleterious variants was loaded at a similar level, the landscape of the variants differed substantially among different populations that two-third of the variants were present in single ethnic populations, and the rest was mostly shared between the populations with closer geographic and genetic relationship. The highly ethnic-specific DDR deleterious variation suggests its potential relationship with different disease susceptibility in ethnic human populations.**

## Introduction

Genetic variation is the driven force of evolution. However, a portion of the genetic variation can be deleterious in causing increased risk of various types of diseases including cancer (Muller, 1950; Eyre-Walker & Keightley, 1999; Lynch, 2010; Xue et al, 2012; Fu et al, 2014; Simons et al, 2014). Different human populations have different susceptibility to diseases, and differential deleterious variation in human populations is considered as a factor contributing to the phenomenon (Kimura et al, 1963; Lohmueller, 2014; Henn et al, 2015). Although this concept is attractive in explaining the relationship between deleterious variants and diseases, the evidence was largely indirect as they were mostly based on the deleterious variants predicted by in silico tools, which is well determined as tending to overpredict the deleteriousness of genetic variants (Richards et al, 2015; Cubuk et al, 2021). In the studies that used the deleterious variants identified from human origin, the results were often restricted by the limited data quantity (Fu et al, 2013), restricted population size (Lohmueller, 2014), or limited to the

populations with specific diseases (Huang et al, 2018). Therefore, it remains largely unclear for the distribution patterns of deleterious variants in human populations.

A genome is constantly attacked by environmental and metabolic factors. The damaged DNA must be fixed timely and spatially to maintain genome stability to avoid pathogenic consequences. Organisms are equipped with a DNA damage repair (DDR) system to repair the damaged DNA. Eukaryotic DDR system consists of at least nine different DDR pathways (Wood et al, 2005; Chatterjee & Walker, 2017). Each DDR pathway contains a group of genes working coordinately to repair a specific type of DNA damage: base excision repair (BER) pathway repairs small, non-helix–distorting base lesions; direct reversal (DR) repair pathway repairs the DNA damaged by ubiquitous alkylating agents; fanconi anemia (FA) pathway repairs the strand cross-link errors; mismatch repair (MMR) pathway repairs mismatch errors; homologous recombination (HR) and nonhomologous end joining (NHEJ) pathways repair double strand breaks; nucleotide excision repair (NER) pathway repairs helix-distorting DNA lesions. However, many DDR genes are prone to germline variation, a part of which can be deleterious in causing increased risk of various diseases including cancer. For example, deleterious variation in BRCA1 of homologous recombination pathway causes high risk of breast and ovarian cancer (Levy-Lahad & Friedman, 2007). Because of their medical value, deleterious variants in human DDR genes have been studied in great detail at the population level and widely used in clinical applications (Wen & Feng, 2004; Milanowska et al, 2011a, 2011b; Spurdle et al, 2012; Knijnenburg et al, 2018).

In this study, we used deleterious variants in DDR genes as a model to study deleterious variation in human populations. We performed an extensive data mining to identify the deleterious variants in DDR genes from 16 human ethnic populations. Comparing these "real-world" data between populations showed substantially different spectrum of DDR deleterious variation among human ethnic populations, although quantitatively the variants were loaded at similar levels. The results highlight that the highly ethnic-specific deleterious variants in DDR genes may contribute to different disease susceptibility in different human ethnic populations.

Cancer Centre and Institute of Translational Medicine, Ministry of Education Frontiers Science Center for Precision Oncology, Faculty of Health Sciences, University of Macau, Macau, China

Correspondence: sanmingwang@um.edu.mo

# Results

## DDR deleterious variation in human populations

We performed genomic data analysis to identify genetic variants in DDR genes. In medical term, "pathogenic" is often used in referring to the genetic variants that contribute to disease and have clinical implications, whereas in biological term, "deleterious" is commonly used in referring to the genetic variants that reduce fitness under purifying selection (MacArthur et al, 2014). In our study, we used "deleterious" instead of "pathogenic," as our study focused on the general populations rather than disease populations.

The nine DDR pathways contain a total of 169 distinct DDR genes based on KEGG and Human DNA Repair Genes databases, in which FA pathway has the highest of 49 DDR genes and directed reversal repair pathway has the lowest of three DDR genes. Through extensive data mining from different sources, we identified 778,723 distinct variants in the 169 distinct DDR genes derived from 158,612 non-disease individuals of 16 ethnic populations. From these variants, we identified 1,781 deleterious variants in 81 DDR genes (47.9% of 169 DDR genes) in eight DDR pathways, but none existed in the three genes in the Direct Reversal pathway (Tables 1 and S1). A database "dbDDR-global" was constructed to host the detailed information for the identified variants (https://genemutation.fhs.um.edu.mo/dbddr-global/). The DDR deleterious variants had the following common features:

(i) Most of the deleterious variants had minor allele frequency (MAF) < 0.001 (1,629 of 1,781 [91.5%]) (Fig 1).
(ii) There were significant differences of deleterious variant–affected genes among different DDR pathways. FA pathway had the highest deleterious variants (926 in 30 of 49 [61%] genes), followed by HR pathway (916 in 21 of 37 [57%] genes), and MMR pathway (188 in 8 of 20 [36%] genes) (Table 1 and Fig 1).
(iii) *BRCA2* had the largest number of deleterious variants (196 of 1,781 total variants, 11.0%), followed by *ATM* (171, 9.6%) and *BRCA1* (126, 7.1%) (Fig 1 and Table S2). This likely reflects their large size (1,863 residues in BRCA1, 3,418 residues in BRCA2, and 3,056 residues in ATM) rather than their high variation frequency.
(iv) The most frequent molecular consequence of deleterious variants was frameshift (39.9%), followed by stop gained (29.6%) and missense variant (11.2%) (Table 2).

## Load of DDR deleterious variants in human populations

We analyzed the quantitative distribution of deleterious variants in the 16 populations (Tables 3 and S3). With 1,781 deleterious variants in the 158,612 individuals included in the study, the average frequency of deleterious variant load was 1.12% in the entire tested populations. In the populations of Japanese (JPN), South Asian (SAS), Chinese (CHN), Korean (KOR), Other East Asian (OEA), Latino/Admixed-American (AMR), African/African American (AFR), Southern European (NFE-SEU), Other non-Finnish European (NFE-ONF), North-Western European (NFE-NWE), and Swedish (NFE-SWE), the load was within twofolds centered at 21 (17–34, Mean ± SD = 21 ± 5.0, Mean ± SE = 21 ± 1.5) per 1,000 individuals (Table 3). However, the load in the populations of Bulgarian (NFE-BGR), Ashkenazi Jewish (ASJ), Finnish (FIN), Icelander (ICE), and Estonia (NFE-EST) varied substantially: Bulgarian had the highest load of 48 per 1,000 individuals, whereas Ashkenazi Jewish, Finnish, Icelander, and Estonia had much lower loads of 11, 7, 2, and 2 per 1,000 individuals, respectively (Table S4). The difference between Bulgarian and Icelander/Estonian reached 19.2-folds. Except the Bulgarian population, the load on these populations was significantly lower than other populations ($P$ = 0.0001).

## Spectrum of DDR deleterious variants in human populations

We compared the spectrum of DDR deleterious variants between the 16 human populations. Of the 1,781 deleterious variants, 1,195 (67%) were present only in single populations (Fig 2A and Table S3). For example, 265 of 322 deleterious variants (82%) in *BRCA1/BRCA2*

**Table 1.** Summary of DNA damage repair (DDR) deleterious variants in DDR pathways

| DDR pathways | Number of genes | Gene with variants (%) | #Variants | Variants/gene | Benjamini–Hochberg[b] |
|---|---|---|---|---|---|
| Homologous Recombination | 37 | 21 (57) | 916 | 44 | 0.433 |
| Fanconi anemia pathway | 49 | 30 (61) | 926 | 31 | 0.103 |
| Mismatch Repair | 20 | 8 (36) | 188 | 24 | 0.672 |
| Nonhomologous end joining | 13 | 7 (54) | 129 | 18 | 0.876 |
| DNA damage response | 15 | 5 (33) | 86 | 17 | 0.450 |
| Nucleotide excision repair | 41 | 13 (32) | 163 | 13 | 0.090 |
| Base excision repair | 32 | 7 (22) | 72 | 10 | **0.020** |
| DNA replication | 34 | 11 (32) | 36 | 3 | 0.130 |
| Direct reversal | 3 | 0 (0) | 0 | 0 | 0.433 |
| Total[a] | 169 | 81 (48) | 1,781 | 17 | **0.020** |

[a]Distinct numbers.
[b]Bold: Statistic significant between pathways.

**Table 2. Molecular consequences of DNA damage repair deleterious variants**

| Molecular consequences | No. | % |
| --- | --- | --- |
| Frameshift variant | 711 | 39.9 |
| Stop gained | 527 | 29.6 |
| Missense variant | 200 | 11.2 |
| Splice donor variant | 139 | 7.8 |
| Splice acceptor variant | 128 | 7.2 |
| Splice region variant | 96 | 5.4 |
| Intron variant | 43 | 2.4 |
| Inframe deletion | 18 | 1.0 |
| Start lost | 17 | 1.0 |
| Synonymous variant | 14 | 0.8 |
| Coding sequence variant | 7 | 0.4 |
| 3 prime UTR variant | 4 | 0.2 |
| Inframe insertion | 1 | 0.1 |
| Total[a] | 1,781 | 100 |

[a]Distinct numbers.

(*BRCA*) and 119 of the 162 deleterious variants (73%) in MMR genes were present in single populations (Table S5).

Of the 586 deleterious variants (23% of the 1,781 variants) shared between populations, 321 (54.8%) were shared between two populations, 120 (20.4%) between three populations, 125 (21.3%) between four and six populations, and only 20 variants (3.4%) over seven populations (Fig 2A and Table S3). The sharing rates were significantly different among the 14 populations except NFE-NWE and NFE-ONF (Table 4). The populations sharing the same deleterious variants tended to be these within nearby geographic regions, such as the Eastern Asian populations (OEA:CHN [49.4%], KOR:JPN [48.6%], JPN:KOR [37.8%], CHN:SAS [36.9%]), and European populations (NFE-BGR:NFE-ONF [67.3%], NFE-ONF:NFE-NEW [66.5%], and NFE-SEU:NFE-NWE [59.3%]) (Table 4). Of all shared variants, only 6.3% were shared with Africa population (Fig 2B). The highly shared deleterious variants where included in *LIG4*, *MUTYH*, *RAD50*, *MSH6*, *OGG1*, *XRCC4*, *ERCC3*, *FANCM*, etc. (Tables 5 and S3). *LIG4* (c.1271_1275del, p.Lys424ArgfsTer20) was shared within 13 populations of CHN, JPN, KOR, SAS, EAS-OEA, ICE, AFR, AMR, NFE-BGR, NFE-NWE, NFE-SEU, NFE-SWE, NFE-ONF except ASJ, NFE-EST, and FIN; *MUTYH* (c.1103G>A, p.Gly368Asp) and *RAD50* (c.2165dup, p.Glu723GlyfsTer5) shared in 12 populations; *MSH6* (c.3226C>T, p.Arg1076Cys) shared in 11 populations; *MUTYH* (c.452A>G, p.Tyr151Cys), *OGG1* (c.137G>A, p.Arg46Gln), and *XRCC4* (c.25del, p.His9ThrfsTer8) shared in 10 populations, *ERCC3* (c.325C>T, p.Arg109Ter), *MSH6* (c.3261dup, p.Phe1088LeufsTer5), and *FANCM* (c.5101C>T, p.Gln1701Ter) shared in nine populations.

The deleterious variants in the populations of Bulgarian, Ashkenazi Jewish, Finnish, Estonia, and Icelander had unique features. Bulgarian population had a higher number of deleterious variants in *ATM* and *MUTYH*; Ashkenazi Jewish population contained the three well-known *BRCA* founder mutations [*BRCA1* 185delAG(c.68_69del), 5382insC(c.5266dup), and *BRCA2* 6174delT(c.5946del)] (Abeliovich et al, 1997); of the only six deleterious variants in Estonian population, two were in *ATM*; *BRCA2* c.771_775del (999del5), a founder mutation in Icelander for breast cancer (Tulinius et al, 2002), was not

but *BRCA2* c.8904del was present in the Icelander population. In the 28 deleterious variants in Icelander population, five were in *TP53*, of which four were only present in Icelander population (Table S4).

### DDR deleterious variants and genetic diseases

We compared the DDR deleterious variation-associated diseases and observed that of the 80 diseases confirmed by mutated DDR genes, 53 (66.3%) are autosomal recessive, 20 (25%) are autosomal dominant, and 7 (8.8%) are both autosomal recessive and dominant (Table S6). For example, Fanconi Anemia caused by 12 mutated DDR genes of *FANCA*, *FANCC*, *FANCD2*, *FANCE*, *FANCF*, *FANCG*, *FANCI*, *FANCL*, *RAD51C*, *SLX4*, *XRCC2*, and *ERCC4* are all autosomal recessive, whereas breast cancer caused by eight DDR genes of *BARD1*, *BRCA1*, *BRCA2*, *BRIP1*, *CHEK2*, *PALB2*, *RAD51D*, and *RAD54L* are all autosomal dominant.

## Discussion

Data from our study provide two important observations: (1) DDR deleterious variants were loaded at similar levels in human populations centered at 21 per 1,000 individuals. As deleterious variants can cause genome instability, they must be present at a tolerable threshold under tight evolution selection pressure. Exceptions were the populations with smaller size or unique evolution history. (2) DDR deleterious variants in human populations were highly ethnic specific. This reflects the genetic diversity of human populations from their adaptation to their natural environments.

Two third of DDR deleterious variants were present only in single ethnic populations. It suggests that DDR deleterious variants could be most likely arisen in recent history (Keinan & Clark, 2012; Fu et al, 2013; Li et al, 2022). This is evidenced by the fact that nearly all currently known *BRCA* founder mutations determined by haplotyping were young, for example, *BRCA1* c.3228_3229del in Italian was arisen 3,225 yr ago (Laitman et al, 2013); of the three *BRCA* founder mutations in Ashkenazi Jewish population, *BRCA1* 185delAG(c.68_69del) was arose 1,500–750 yr ago (Hamel et al, 2011), *BRCA1* 5382insC(c.5266dup) 1,800 yr ago (Neuhausen et al, 1998), and *BRCA2* 6174delT(c.5946del) 580 yr ago (Zeegers et al, 2004); *BRCA2* c.9118-2A>G, a founder mutation in Icelander population, was arisen only 220–144 yr ago (Altmann & Gennery, 2016). Our recent study also revealed that human *BRCA* deleterious variants mostly arose after migration out-of-Africa and great expansion of modern human population (Li et al, 2022). This may also be related with differences of evolution selection on different DDR genes. For example, *BRCA* is under strong positive selection, but MMR is under negative/neutral selection (Zhang et al, 2021). This contributed to more *BRCA* deleterious variants than MMR deleterious variants, as reflected by the 1.2% of *BRCA* deleterious variants shared between non-African and African populations whereas 9.9% sharing rate in MMR deleterious variants (Fig 2C and D).

A third of DDR deleterious variants were shared mostly between geographically related populations. The penetrance of the highly shared deleterious variants can be lower in causing phenotype change. For example, *LIG4* (c.1271_1275del, p.Lys424ArgfsTer20) was shared in 13 populations. *LIG4* is a member in nonhomologous end joining pathway. While mutation in *LIG4* can cause autosomal recessive diseases of immune deficiency, growth failure, sensitive to ionizing radiation, and cancer (Altmann & Gennery, 2016; Taskiran et

**Table 3. Number of DNA damage repair deleterious variants identified in different ethnic populations**

| Ethnic population | Abbreviation | Number of individuals | #Variants | Load ($P$ = 0.001)[a] |
|---|---|---|---|---|
| Bulgarian | NFE-BGR | 1,335 | 64 | 48 |
| Southern European | NFE-SEU | 5,805 | 198 | 34 |
| Other non-Finnish European | NFE-ONF | 16,568 | 420 | 25 |
| Japanese | JPN | 3,552 | 79 | 22 |
| North-Western European | NFE-NWE | 25,410 | 544 | 21 |
| Chinese | CHN | 10,588 | 216 | 20 |
| South Asian | SAS | 15,263 | 305 | 20 |
| Korean | KOR | 2,964 | 57 | 19 |
| Swedish | NFE-SWE | 13,067 | 244 | 19 |
| Latino/Admixed American | AMR | 17,554 | 312 | 18 |
| African/African American | AFR | 11,810 | 202 | 17 |
| Other East Asian | EAS-OEA | 7,992 | 133 | 17 |
| Ashkenazi Jewish | ASJ | 4,931 | 56 | 11 |
| Finnish | FIN | 12,554 | 93 | 7 |
| Estonian | NFE-EST | 2,418 | 6 | 2 |
| Icelander | ICE | 12,584 | 27 | 2 |
| Total[b] | | 158,612 | 1,781 | 11 |

[a]Load = variants/individuals*1,000 ($P$.value between group 34-17 and group 11-2).
[b]Distinct number.

al, 2019), only 36 cases of diseases caused by *LIG4* mutation had been reported so far (Taskiran et al, 2019). It would be interesting to know if there could be any beneficial impact for these commonly shared deleterious variants, similar to the hemoglobin S and C variants in conferring resistance to malaria infection (Ha et al, 2019).

The load of DDR deleterious variants in Ashkenazi Jewish, Finnish, Icelander, and Estonia was much lower than in other populations. Each of these populations had its unique evolution history. For example, the initial population sizes were small in Ashkenazi Jewish (Guha et al, 2012), Icelander (Andersen & Zoega, 1999), Finnish (Kere, 2001), Estonia (Pankratov et al, 2020). Therefore, their population structures could be affected by the effects of bottleneck, founder and genetic drift (Crow, 1970). The small founder individuals and genetic isolation contributed to the unique genetic features of Finnish population in distinguishing them from other European populations (Harris, 2015; Kerminen et al, 2017). It is unlikely that the limited sample size of these populations included in the study contributed to their low detection of deleterious variants, as nearly four-times more deleterious variants were identified in Bulgarian than in Estonian, although the sample size of Bulgarian was 1.8-fold smaller than that of Estonian. Ashkenazi Jewish population has its unique types of genetic defect–contributed diseases (https://www.jewishvirtuallibrary.org/ashkenazi-jewish-genetic-diseases). For example, the three *BRCA* founder mutations [*BRCA1* 185delAG (c.68_69del), 5382insC (c.5266dup), and *BRCA2* 6174delT (c.5946del)] have high carrier frequency (2.17%) in Ashkenazi Jewish population contributing to high risk of breast and ovarian cancer (Gabai-Kapara et al, 2014). Although *BRCA2* c.771_775del (999del5) is the major founder mutation in Icelander breast cancer (Thorlacius et al, 1997), it was not present in the 27 DDR deleterious variants

identified in the Icelander population of 12,584 individuals included in our study. Its absence in Icelander general population highlights the possibility that *BRCA2* c.771_775del (999del5) may have lower prevalence in Icelander general population but be enriched in Icelander breast cancer cohort (Tulinius et al, 2002).

It is particularly interesting that most of the diseases caused by the mutated DDR genes are autosomal recessive. This can substantially diminish the impact of the DDR deleterious variants in disease susceptibility in human population although the prevalence can be high, as reflected by the rarity of the diseases caused by autosomal recessive *LIG4* deleterious variation, whereas the impact could be higher in the populations with consanguinity culture (Bittles & Black, 2010). It is also interesting to note that the deleterious variants in certain DDR genes causing autosomal dominant diseases can also be highly prevalent in human populations. This is represented by the high cancer-risk deleterious variants in *BRCA1* and *BRCA2* that the carrier rate reaches to one in a few hundreds of individuals in general population, for example, one in 384 in Japanese population (Momozawa et al, 2018), one in 265 in Chinese Han and Mexican populations (Fernández-Lopez et al, 2019; Dong et al, 2021), one in 256 in Malaysian population (Wen et al, 2018), one in 189 in US population (Manickam et al, 2018), and the highest of one in 46 in Ashkenazi Jewish population (Gabai-Kapara et al, 2014). Besides their deleterious effects, there could be beneficial significance for the high prevalent high-risk genetic predisposition in human population. In contrast to the stable status in most species, human *BRCA* is under strong positive selection leading to its high variability of more than 70,000 variants identified so far (Huttley et al, 2000; Cline et al, 2018). Besides the classical function of DDR, *BRCA*

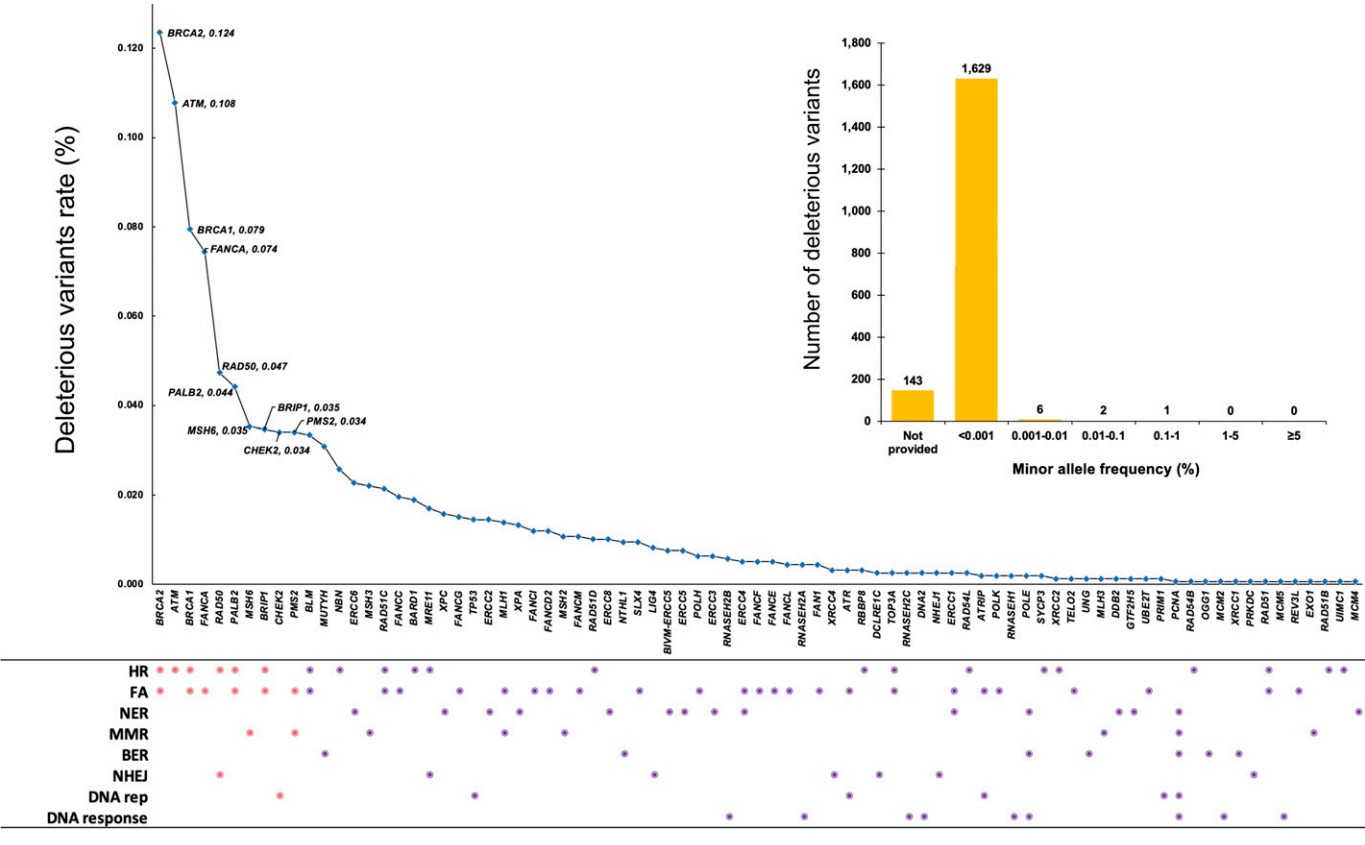

**Figure 1. Frequency of deleterious variant distribution in DNA damage repair (DDR) genes.**
It shows the distribution frequency of the deleterious variants in 81 DDR genes in the 159,612 individuals included in the study. The dots in DDR pathways show the gene(s) in DDR pathways affected by the variants. Pink dot refers to the high frequent variant-affected top 10 DDR genes of *BRCA2, ATM, BRCA1, FANCA, RAD50, PALB2, MSH6, BRIP1, CHEK2,* and *PMS2* and their corresponding pathways. BAR chart shows the distribution of minor allele frequency (%) for the 1,781 deleterious variants. HR, homologous recombination; FA, fanconi anemia; NER, nucleotide excision repair; MMR, mismatch repair; BER, base excison repair; NHEJ, non-homologous end joining; DNA rep, DNA replication; DNA response, DNA damage response.

gains multiple new functions including regulation of immunity against viral infection (Lou et al, 2014) and gene expression (Rosen et al, 2006), promotion of neural development (Pao et al, 2014), and enhancement of reproduction (Smith et al, 2013).

A limitation of our study is the lack of sufficient DDR data from non-European populations. It reinforces the importance of studying diverse populations in human genetic study (Sirugo et al, 2019; Sakaue et al, 2021).

Our study focused on the deleterious variation in DDR genes, which are only a part of the genes with deleterious effects. It will be interesting to know what we observed in DDR deleterious variation could also be present to the genes of other functional categories in human populations. It will also be interesting to know if the differences of deleterious variation may be linked to different susceptibility of human populations to diseases.

# Materials and Methods

### Sources of deleterious variation data

The DDR genes were determined by combining the genes from the "Replication and repair" in KEGG (Kanehisa & Goto, 2000, https://www.genome.jp/kegg/pathway.html#cellular) and the Human DNA Repair Genes (Wood et al, 2005, https://www.mdanderson.org/

documents/Labs/Wood-Laboratory/human-dna-repair-genes.html). Genetic variants in DDR genes of general human populations were collected from the following resources: Chinese population from the China Metabolic Analytics Project (ChinaMAP) (Cao et al, 2020, http://www.mbiobank.com/, accessed in 9 September 2020); Japanese population from the 3.5KJPNv2 (Tadaka et al, 2019; https://www.megabank.tohoku.ac.jp/english/about-the-change-on-the-release-of-3-5kjpn/, accessed 23 September 2020); Korean population from the Korean Variant Archive (KOVA) (Lee et al, 2017, http://kobic.re.kr/kova/, accessed 29 September 2020) and gnomADv2 noncancer data (Lee et al, 2017; Karczewski et al, 2020; https://gnomad.broadinstitute.org/, accessed 16 December 2020); Icelander population from the deCODE (https://www.ebi.ac.uk/eva/?eva-study=PRJEB15197, accessed 1 July 2020) after filtered by the variant data from Icelander patients (Gudbjartsson et al, 2015; Jonsson et al, 2017; https://www.ebi.ac.uk/eva/?eva-study=PRJEB8636, accessed 26 September 2020); variation data of non-Finnish European (Estonian, Bulgarian, Swedish, Southern European, North-Western European and Other Non-Finnish European), Finnish, Latino/Admixed-American, Ashkenazi Jewish, African/African American, South Asian, other East Asian were extracted from gnomADv2 noncancer data. In each of the original studies, ancestry for each population was tested by either principal component analysis (Chinese, Japanese, Korea, gnomADv2) or genotyping (Icelander) as indicated in the original studies. Whole genome sequence data were from ChinaMAP, Japanese 3.5KJPNv2, Icelander deCODE, whole

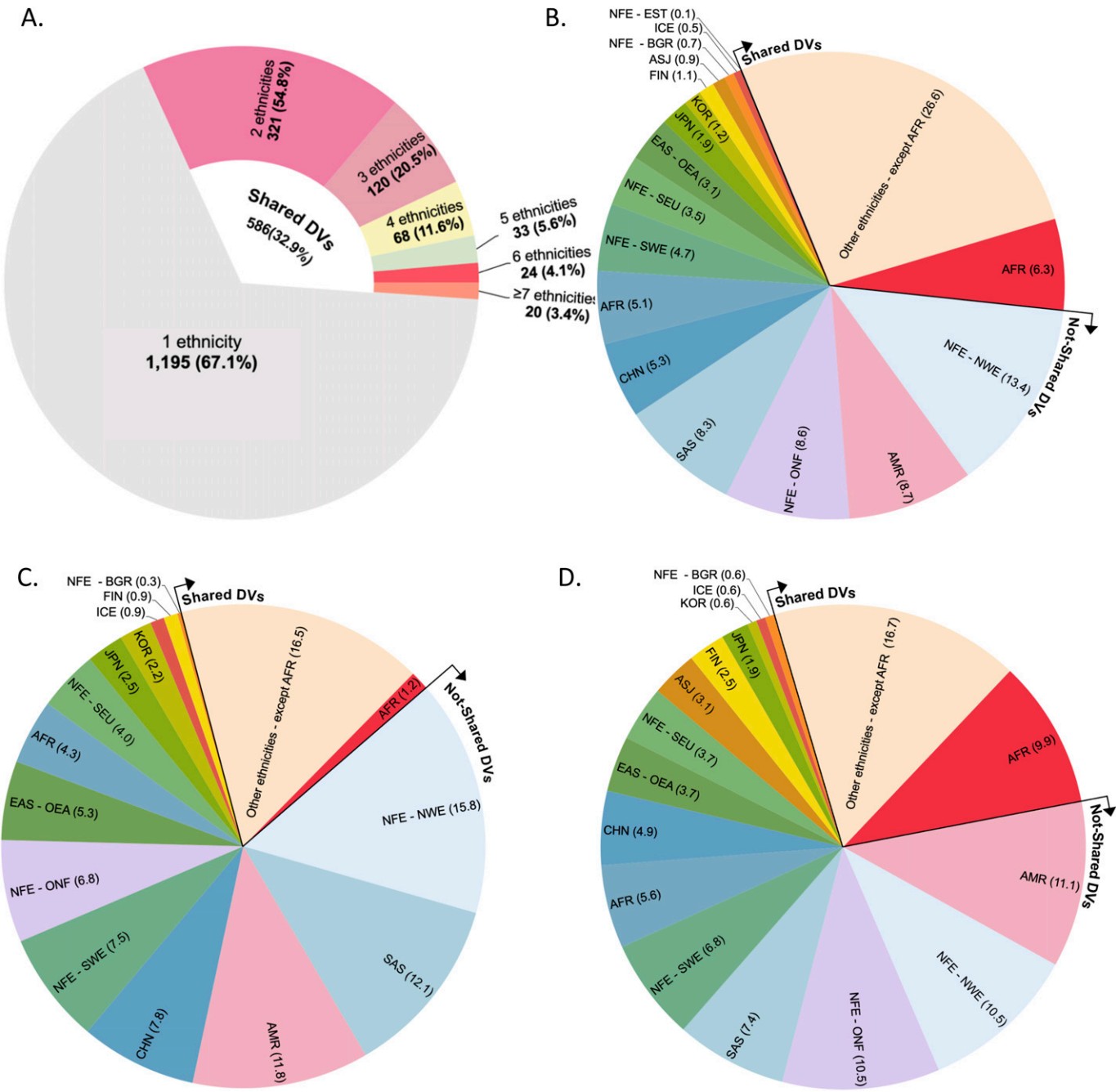

**Figure 2. DNA damage repair deleterious variants distributed in human populations.**
**(A)** Ethnic specificity of DNA damage repair deleterious variants. It shows that 1,195 of the 1,781 variants were present in single populations, and the rest were shared mostly between two populations. **(B)** DDR variants sharing between non-Africa and African populations. **(C)** *BRCA1/2* variant deleterious variants sharing between different populations. **(D)** MMR variants sharing between different populations. The different sharing rates between *BRCA* and MMR variants showed the more variable *BRCA* deleterious variants than MMR deleterious variants. DV, deleterious variants.

exome sequence data were from Korean KOVA, whole genome sequence and whole exome sequence data were from gnomADv2. Only the variants marked as "PASS" in the corresponding VCF file were used in our study. The genome position of variants was based on human reference genome sequences hg38.

We used ANNOVAR program to annotate the variants (Wang et al, 2010), including annotation of the genetic information by referring to refGene, dbSNP150 and COSMIC database, and annotation of the MAF by referring to gnomAD, ExAC, ESP6500, and 1,000 genomes. Ensembl Variant Effect Predictor was used to annotate the molecular consequence of each variant. "intron variant," "upstream gene variants," "downstream gene variant," "5'UTR variant," and "3'UTR variant" were grouped as non-coding variants; "missense variant," "synonymous variant," "frameshift variants," "inframe deletion," "start lost,"

**Table 4. Comparison of DNA damage repair deleterious variants among 16 ethnic populations**

| Ethnicity | Total | Unshared (%) | Shared (%)[a] | | | | | | | | | | | | | | | | | | Ave. | Benjamini–Hochberg |
|---|---|---|---|---|---|---|---|---|---|---|---|---|---|---|---|---|---|---|---|---|---|---|
| | | | Total | CHN | JPN | KOR | EAS-OEA | SAS | ICE | AFR | AMR | ASJ | FIN | NFE-BGR | NFE-EST | NFE-SEU | NFE-SWE | NFE-NWE | NFE-ONF | | |
| CHN | 216 | 94 (44) | 122 | – | 12.3 | 9.0 | 31.1 | **36.9** | 1.6 | 25.4 | 29.5 | 6.6 | 12.3 | 5.7 | 0.8 | 25.4 | 20.5 | 36.1 | 29.5 | 17.7 | **2.74E-03** |
| JPN | 79 | 34 (43) | 45 | 33.3 | – | **37.8** | 13.3 | 31.1 | 2.2 | 15.6 | 17.8 | 2.2 | 0.0 | 2.2 | 0.0 | 8.9 | 17.8 | 22.2 | 15.6 | 13.8 | **7.52E-04** |
| KOR | 57 | 22 (39) | 35 | 31.4 | **48.6** | – | 14.3 | 14.3 | 2.9 | 20.0 | 20.0 | 2.9 | 5.7 | 5.7 | 0.0 | 14.3 | 11.4 | 11.4 | 17.1 | 13.8 | **7.52E-04** |
| EAS-OEA | 133 | 56 (42) | 77 | **49.4** | 7.8 | 6.5 | – | 26.0 | 1.3 | 19.5 | 16.9 | 5.2 | 9.1 | 6.5 | 1.3 | 18.2 | 16.9 | 42.9 | 36.4 | 16.5 | **7.52E-04** |
| SAS | 305 | 148 (49) | 157 | 28.7 | 8.9 | 3.2 | 12.7 | – | 3.2 | 28 | 36.3 | 6.4 | 11.5 | 9.6 | 0.0 | 25.5 | 25.5 | **47.1** | 35.0 | 17.6 | **2.13E-03** |
| ICE | 27 | 9 (33) | 18 | 11.1 | 5.6 | 5.6 | 5.6 | 27.8 | – | 33.3 | 50 | 5.6 | 22.2 | 16.7 | 5.6 | 38.9 | 38.9 | **55.6** | **55.6** | 23.6 | **7.31E-04** |
| AFR | 202 | 90 (45) | 112 | 27.7 | 6.2 | 6.2 | 13.4 | 39.3 | 5.4 | – | 33.9 | 5.4 | 13.4 | 12.5 | 0.0 | 24.1 | 27.7 | **50.0** | 43.8 | 19.3 | **8.69E-04** |
| AMR | 312 | 155 (50) | 157 | 22.9 | 5.1 | 4.5 | 8.3 | 36.3 | 5.7 | 24.2 | – | 8.9 | 15.3 | 12.7 | 0.6 | 35.7 | 24.2 | **53.5** | 45.2 | 18.9 | **1.63E-03** |
| ASJ | 56 | 16 (29) | 40 | 20.0 | 2.5 | 2.5 | 10.0 | 25.0 | 2.5 | 15.0 | 35.0 | – | 10.0 | 27.5 | 0.0 | 42.5 | 22.5 | **62.5** | **62.5** | 21.2 | **7.31E-04** |
| FIN | 93 | 19 (20) | 74 | 20.3 | 0.0 | 2.7 | 9.5 | 24.3 | 5.4 | 20.3 | 32.4 | 5.4 | – | 21.6 | 1.4 | 33.8 | **52.7** | **52.7** | **52.7** | 20.9 | **7.52E-04** |
| NFE-BGR | 64 | 12 (19) | 52 | 13.5 | 1.9 | 3.8 | 9.6 | 28.8 | 5.8 | 26.9 | 38.5 | 21.2 | 30.8 | – | 0.0 | 53.8 | 48.1 | 59.6 | **67.3** | 25.6 | **7.52E-04** |
| NFE-EST | 6 | 2 (33) | 4 | 25.0 | 0.0 | 0.0 | 25.0 | 0.0 | 25.0 | 0.0 | 25.0 | 0.0 | 25.0 | 0.0 | – | **50.0** | 25.0 | **50.0** | **50.0** | 18.8 | **7.31E-04** |
| NFE-SEU | 198 | 63 (32) | 135 | 23.0 | 3.0 | 3.7 | 10.4 | 29.6 | 5.2 | 20 | 41.5 | 12.6 | 18.5 | 20.7 | 1.5 | – | 33.3 | **59.3** | 54.8 | 21.1 | **1.10E-03** |
| NFE-SWE | 244 | 83 (34) | 161 | 15.5 | 5.0 | 2.5 | 8.1 | 24.8 | 4.3 | 19.3 | 23.6 | 5.6 | 24.2 | 15.5 | 0.6 | 28.0 | – | 54.0 | **54.7** | 17.9 | **8.69E-04** |
| NFE-NWE | 544 | 238 (44) | 306 | 14.4 | 3.3 | 1.3 | 10.8 | 24.2 | 3.3 | 18.3 | 27.5 | 8.2 | 12.7 | 10.1 | 0.7 | 26.1 | 28.4 | – | **57.8** | 15.4 | 6.14E-01 |
| NFE-ONF | 420 | 154 (37) | 266 | 13.5 | 2.6 | 2.3 | 10.5 | 20.7 | 3.8 | 18.4 | 26.7 | 9.4 | 14.7 | 13.2 | 0.8 | 27.8 | 33.1 | **66.5** | – | 16.5 | 4.12E-01 |
| Total[b] | 1,781 | 1,195 (67) | 583 (33) | | | | | | | | | | | | | | | | | 18.7 | |

[a]% = shared variants/total shared variants × 100; the rate in bold refers to the highest sharing rate among populations.
[b]Distinct number.

**Table 5. Top 10 highly shared DNA damage repair deleterious variants in human populations**

| Gene | HGVSc | HGVSp | Frequency | Disease | Population shared | Number |
|------|-------|-------|-----------|---------|-------------------|--------|
| *LIG4* | c.1271_1275del | p.Lys424ArgfsTer20 | 0.0002 | LIG4-Related disorders | CHN, JPN, ICE, KOR, AFR, AMR, EAS_OEA, NFE_BGR, NFE_NWE, NFE_SEU, NFE_SWE, NFE_ONF, SAS | 13 |
| *MUTYH* | c.1103G>A | p.Gly368Asp | 0.0030 | MYH-associated_polyposis | CHN, AFR, AMR, ASJ, EAS_OEA, FIN, NFE_BGR, NFE_NWE, NFE_SEU, NFE_SWE, NFE_ONF, SAS | 12 |
| *RAD50* | c.2165dup | p.Glu723GlyfsTer5 | 0.0003 | Hereditary cancer | CHN, AFR, AMR, ASJ, EAS_OEA, FIN, NFE_BGR, NFE_NWE, NFE_SEU, NFE_SWE, NFE_ONF, SAS | 12 |
| *MSH6* | c.3226C>T | p.Arg1076Cys | 0.0001 | Lynch syndrome | CHN, AFR, AMR, ASJ, EAS_OEA, NFE_BGR, NFE_NWE, NFE_SEU, NFE_SWE, NFE_ONF, SAS | 11 |
| *MUTYH* | c.452A>G | p.Tyr151Cys | 0.0015 | MYH-associated polyposis | CHN, AFR, AMR, FIN, NFE_BGR, NFE_NWE, NFE_SEU, NFE_SWE, NFE_ONF, SAS | 10 |
| *OGG1* | c.137G>A | p.Arg46Gln | 0.0022 | Clear cell carcinoma of kidney | AFR, AMR, ASJ, FIN, NFE_BGR, NFE_NWE, NFE_SEU, NFE_SWE, NFE_ONF, SAS | 10 |
| *XRCC4* | c.25del | p.His9ThrfsTer8 | 0.0004 | Short stature | ICE, AFR, AMR, FIN, NFE_BGR, NFE_NWE, NFE_SEU, NFE_SWE, NFE_ONF, SAS | 10 |
| *ERCC3* | c.325C>T | p.Arg109Ter | 0.0005 | Unknown | AMR, ASJ, FIN, NFE_BGR, NFE_NWE, NFE_SEU, NFE_SWE, NFE_ONF, SAS | 9 |
| *MSH6* | c.3261dup | p.Phe1088LeufsTer5 | 0.0001 | Lynch syndrome | CHN, ICE, AFR, AMR, FIN, NFE_NWE, NFE_SWE, NFE_ONF, SAS | 9 |
| *FANCM* | c.5101C>T | p.Gln1701Ter | 0.0013 | Fanconi anemia | ICE, AFR, AMR, FIN, NFE_BGR, NFE_NWE, NFE_SEU, NFE_SWE, NFE_ONF | 9 |

"stop gained," "stop retained variant," "splice region variant," "splice donor variant," "splice acceptor variant," "coding sequence variant" and "protein altering variant" were grouped as coding variants. Clinical significance of each variant was classified as Pathogenic, Likely pathogenic, Variants of Uncertain Significance (VUS), Likely benign, and Benign by referring to ClinVar (Landrum et al, 2016; released 1 May 2021, imbedded in ANNOVAR). In our study, we defined the pathogenic and likely pathogenic variants as deleterious variants.

### Construction of DDR deleterious variant database

We developed an open accessing database "dbDDR-GLOBAL" to host the DDR deleterious variants identified in the 16 populations (https://genemutation.fhs.um.edu.mo/dbddr-global/). The database provides detailed information for each variant including genome position, gene name, molecular consequence, classification, SNP ID, MAF, population origin, etc.

### Statistical analysis

Statistical analysis was performed via R program. Chi test ($\chi$) was used to compare the differences between DDR pathways with deleterious variant-affected DDR genes, and double-side $t$ test was used to compare the differences of deleterious variant loads among populations. We further performed Benjamini–Hochberg procedure for chi test ($\chi$) and $t$ test results, $P < 0.05$ was considered as statistically significant.

## Data Availability

The original data used in the study were from public resources as indicated in the text, the resulting data were provided as online Tables S1–S6, and in the database "dbDDR-global" for users to explore the data (https://genemutation.fhs.um.edu.mo/dbddr-global/).

### Expanded view

The online version contains Tables S1–S6.

## Supplementary Information

## Acknowledgements

This work was performed at the high-performance computing cluster supported by Information and Communication Technology Office of the University of Macau. This work was funded by grants from Macau Science and Technology Development Fund (085/2017/A2 and 0077/2019/AMJ), the University of Macau (SRG2017-00097-FHS, MYRG2019-00018-FHS, and MYRG2020-00094-FHS), and the Faculty of Health Sciences, University of Macau (Startup fund, FHSIG/SW/0007/2020P, FHS Innovation grant) (SM Wang).

### Author Contributions

Z Qin: data curation, formal analysis, investigation, visualization, and writing—original draft, review, and editing.
T Huang: resources, software, database construction, and writing—review and editing.
M Guo: data analysis and writing—review and editing.

SM Wang: conceptualization, funding acquisition, investigation, and writing—review and editing.

## Conflict of Interest Statement

The authors declare that they have no conflict of interest.

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
