## [Reviewer comments · Life Science Alliance]

Distinct landscapes of deleterious variants in DNA damage repair system in ethnic human populations

Zixin Qin, Teng Huang, Maoni Guo, San Ming Wang

DOI: 10.26508/lsa.202101319

Corresponding author(s): Prof. San Ming Wang (University of Macau)

Review timeline:

Submission Date:	2021-11-26
Editorial Decision:	2022-01-25
Appeal Requested:	2022-02-11
Editorial Decision:	2022-02-11
Revision Received:	2022-03-29
Editorial Decision:	2022-05-11
Revision Received:	2022-05-11
Accepted:	2022-05-11

Scientific Editor: Novella Guidi

Transaction Report:

Re: Life Science Alliance manuscript #LSA-2021-01319-T

Prof. San Ming Wang
University of Macau
Faculty of Health Sciences
University of Macau
Taipa
Macau SAR, Taipa 999078
Macao

Dear Dr. Wang,

Thank you for submitting your manuscript entitled "Spectrum rather than load of deleterious variants better explains human disease susceptibility". The manuscript has been evaluated by expert reviewers, whose reports are appended below. Unfortunately, after an assessment of the reviewer feedback, our editorial decision is against publication in Life Science Alliance.

Although your manuscript is intriguing, I feel that the points raised by the reviewers are more substantial than can be addressed in a typical revision period. If you wish to expedite publication of the current data, it may be best to pursue publication at another journal.

Given the interest in the topic, I would be open to resubmission to Life Science Alliance of a significantly revised and extended manuscript that fully addresses the reviewers' concerns and is subject to further peer-review. If you would like to resubmit this work to Life Science Alliance, you may submit an appeal directly through our manuscript submission system. Please note that priority and novelty would be reassessed at resubmission.

Regardless of how you choose to proceed, we hope that the comments below will prove constructive as your work progresses. We would be happy to discuss the reviewer comments further once you've had a chance to consider the points raised in this letter.

Thank you for thinking of Life Science Alliance as an appropriate place to publish your work.

Sincerely,

Reviewer #1 (Comments to the Authors (Required)):

Review for Ms. No. LSA-2021-01319-T, Life Science Alliance

In the paper entitled "Spectrum rather than load of deleterious variants better explains human disease susceptibility", Drs Qin, Huang and Wang discuss the distribution of presumably pathogenic variants (altering genes associated with DNA damage repair [DDR]) in populations of different ancestries.

The concepts explored are interesting and important but I have a number of major concerns which prevents recommending publication:

MAJOR ISSUES

1. The main conclusion ("the spectrum rather than the load of DDR deleterious variants is the major contributor to different disease susceptibility in human populations") is not supported by the findings. There is no specific mention or quantitative analysis/data on the differences in prevalence of specific groups of conditions among various populations. For example, it is unclear if the observed pattern of genetic variation in BRCA1 and BRCA2 would fit epidemiological data on the prevalence of breast cancer. In fact, to gain insights into this, information on the minor allele frequency of the different alleles and the mode of inheritance of the various disease-implicated genes need to be taken into account. Overall, I would recommend avoiding overinterpretation of the results and changing the core theme of the study. I feel that something around the "biogeographical patterns of pathogenic variants in DDR-implicated genes" would be a less ambitious but more appropriate angle here.
2. It is counterintuitive that a heterozygous change in a gene associated with dominant disease is treated as equal to a heterozygous change in a gene associated with recessive disease. This needs to be factored in and, as a minimum, a sub-analysis focusing on variants linked to autosomal dominant disease only, should be performed. In fact, it is tempting to adopt a methodology that only looks at this group.
3. The authors need to account for and/or discuss ascertainment bias. It seems that ClinVar was used as a key filter but of course groups from Europe and North America are more likely to deposit data in this repository.

MINOR ISSUES

The article contains a number of typos and imprecise descriptions and would benefit from further proof reading. I have highlighted some of these issues that were encountered in the introduction to point to this issue but highlighting these inaccuracies throughout the manuscript is beyond the scope of this review.

INTRODUCTION

- ** replace "the phenonium" with "this phenomenon"
- ** "it has seldomly been tested by the actual deleterious data from human populations." (unclear/imprecise phrasing - please amend)
- ** replace "existing evidences" with "existing evidence"
- ** replace "current arts" with "current tools"
- ** replace "questionable that they tend" with "questionable as they tend"
- ** "Certain studies indeed used the actual deleterious variants derived from human origin." (unclear/imprecise phrasing - please amend).
- ** "However, many DDR genes themselves are prone to germline variation, in which a part is deleterious leading to high risk of diseases" (unclear/imprecise phrasing - please amend).
- ** replace "Because of their medical values" with "Because of their medical value".
- ** replace "have been studied in very details" with "have been studied in great detail".
- ** replace "characterized at the populational level" with "characterized at the population level".
- ** "Taking the advantages, we used the deleterious variation data in human DDR genes as a model to test the concept of deleterious load in human populations." (unclear/imprecise phrasing - please amend).
- ** "highlighting that ethnic-specific deleterious variants rather than the loading is the major determinant for the differential disease susceptibility in human populations."

(unclear/imprecise phrasing - please amend)

RESULTS

** "The 9 DDR pathways contain a total of 276 DDR genes composed of 169 distinct DDR genes": it is unclear whether the number of genes is 276 or 169. Please amend/clarify.

** the term "deleterious" is used in the introduction while the term "pathogenic" is predominantly used in the results. The authors should use consistent terminology, provide definitions and justify their choice of terms. For example, as per MacArthur et al (Nature 2014, DOI: 10.1038/nature13127) deleterious is a variant that reduces the reproductive fitness of carriers (and would thus be targeted by purifying natural selection). In the same paper, the term pathogenic is used to define a variant that contributes mechanistically to disease, but is not necessarily fully penetrant (i.e., may not be sufficient in isolation to cause disease). I propose that the authors introduce and consistently use the term "presumed pathogenic" or "presumed functional" to describe the group of variants that they considered.

** In the methods, it seems that Clinvar was the primary source used for assigning functional significance to variants; it would be useful to include a mention to this in the results. Similarly, a mention to the source of the population data would enhance the readability of the manuscript.

** the term "somatic pathogenic variants" implies that the changes were not detected in the germline. It would be clearer if the description "presumed pathogenic, COSMIC-listed variants" was used instead.

DISCUSSION

** "In contrast to the widely belied view, the results from our current study provide evidence to show that the load of deleterious variants is at similar level in different human populations due possibly to a tolerating threshold controlled under tight evolution selection, whereas the spectrum of deleterious variants is highly ethnic-specific in reflecting human adaptation to different local environments". (unclear/imprecise phrasing with a tendency to overinterpret the results; please amend and include references when discussing the widely accepted view).

** "Therefore, the different susceptibility of human populations to disease as represented by DDR deleterious variants can be better explained by the differences of deleterious variant spectrum than the different level of deleterious variant load." (the authors did not study the correlation between variation and prevalence of disease, thus the above conclusion is not substantiated by the study results [see also Major issues section]).

** A mention to the age of genetic variation is included in the third paragraph of the discussion. Further analysis to look into this important topic could be performed using tools like <https://human.genome.dating/> .

** "The absence of the founder mutation BRCA2 999del5 in Icelander population highlights that it is at lower prevalence level in general population but enriched in Icelander breast cancer cohort" (this is a confusing statement; please clarify and include references; also please use HGVS nomenclature when describing genetic variants).

MATERIALS and METHODS

** More information is required on how the ancestral groups were determined. There is a mention to the 'source of variation data' in the first section of the Methods but it would be worth providing more details (e.g. were all populations defined using PCA-based analysis?)

** "variation data of non-Finnish European (Estonian, Bulgarian, Swedish, Southern European, North-Western European and Other Non-Finnish European)" (please clarify the source of these data - would it be fair to presume that gnomADv2 was used?).

** "Pathogenic and Likely pathogenic variants (PVs) were defined as deleterious

variants and used in the study." (the abbreviation PV is used extensively in the text and should be properly define. It sounds like the authors considered all changes listed as pathogenic or likely pathogenic in Clinvar as functional and included them in a group that they used for further analysis. If that is the case, I would advice using the term "presumed pathogenic" or "presumed functional" to describe this subset of genetic variants.

** " $p < 0.05$ was considered as statistically significant." (it is likely that a degree of multiple testing was involved; how did the authors deal with this important issue?)

Reviewer #2 (Comments to the Authors (Required)):

Review on "Spectrum rather than load of deleterious variants better explains human disease susceptibility" by Qin et al.

This is a short report in which the authors collected DNA sequence information from multiple genomic databases representing 16 ethnic groups, and analyzed the data to evaluate frequencies of "pathological variants" of most of mammalian DNA repair genes. Geographically close ethnic groups (such as Japanese v.s. Korean) show more similarity in the allele distribution than those less related. Some of the ethnic group (e.g., Bulgarian collection) show an unusually high PV frequency compared to other groups.

Major weaknesses are listed.

- The conclusion that PV frequencies resemble between two kin ethnic groups is not a novel finding. In fact, to reach this conclusion, more rigorous way would be to analyze all variants including single nucleotide polymorphisms (i.e., benign variants).
- This report lacks association of the genetic variations with clinical consequences. Is the Bulgarian population predisposed to certain types of cancers due to the high PV? Or does the Ashkenazi Jewish population have lower risks of diseases caused by the DDR genes? It is a little counter intuitive as Ashkenazi Jewish population often is represented with a unique spectrum of genetic diseases.
- Table 1 lists numbers of individuals participated and the numbers of PVs. This is not sufficient for a report, the report should clarify what are the major PV alleles (e.g., BRCA1 truncation mutations) and show their allele frequencies for each ethnic group.

Other comments

- Introduction introduces DDR with "at least 9 different DDR pathways", but the following sentences describes only 7.
- This study collected multiple sources of genetic data. How each institution analyzed the DNA sequencing should be described. For example, are they all based on the whole exome sequencing, whole genome sequencing, or based on RNAseq?
- Page 7, the second paragraph: it discusses the possibility of an advantageous LIG4 allele in an environment, but it is counter intuitive because it also describes LIG4 is one of the genes shared by many ethnic groups. Are these 13 ethnic groups geographically close, or sharing a similar environment that has a health impact in any way?
- There are many grammatical errors including typos and misuse of words. The preposition "between" has been misused at too many places (where "among" is more appropriate). It is strongly recommended to have a professional English editor go through the entire manuscript.

Appeal Request

25 March 2022

Dear Dr. Guidi,

The authors of manuscript #LSA-2021-01319-T have requested an appeal:

We will follow the comments by the reviewers to prepare the revision for resubmission.

Sincerely,

Editorial Staff

Editorial Decision on Appeal Request

11 February 2022

MS: LSA-2021-01319-T

Prof. San Ming Wang
University of Macau
Faculty of Health Sciences
University of Macau
Taipa
Macau SAR, Taipa 999078
Macao

Dear Dr. Wang,

I am happy to hear that you will be addressing all the reviewers' points in your revised version. The link in this letter will allow you to resubmit your revised manuscript once you'll be ready with the revisions.

Yours sincerely,

Reviewer #1**Question**

The main conclusion ("the spectrum rather than the load of DDR deleterious variants is the major contributor to different disease susceptibility in human populations") is not supported by the findings. There is no specific mention or quantitative analysis/data on the differences in prevalence of specific groups of conditions among various populations. For example, it is unclear if the observed pattern of genetic variation in BRCA1 and BRCA2 would fit epidemiological data on the prevalence of breast cancer. In fact, to gain insights into this, information on the minor allele frequency of the different alleles and the mode of inheritance of the various disease-implicated genes need to be taken into account. Overall, I would recommend avoiding overinterpretation of the results and changing the core theme of the study. I feel that something around the "biogeographical patterns of pathogenic variants in DDR-implicated genes" would be a less ambitious but more appropriate angle here.

Reply

Thanks very much for this valuable comment. Indeed, there was no direct data in our study in linking the DDR variation to disease susceptibility. Following the comments, we performed extensive searching to find the potential supportive epidemiological data. We found that the most relevant data are the WHO cancer statistics. However, the data are based on the ICD10 system, which provide the overall data from each cancer type but doesn't separate further into hereditary (germline, around 10% of total cases attributed with germline mutation) or sporadic (somatic, the majority of cancer cases attributed with somatic mutation) cancer. Our study is focused on germline variant, without the actual data from hereditary cancer, it is difficult to link the data to the DDR data in different populations. We fully agree with Reviewer's comments that it is too ambitious for the study to link genotype with phenotype in human populations without actual evidence as it overinterpretes the data to disease susceptibility, instead to put the focus of the study on the general patterns of DDR variation in human populations. Regarding the title of the manuscript, we propose to use "Distinct landscapes of deleterious variants in DNA damage repair system in ethnic human populations", which addresses the difference of variant spectrum in different ethnic population.

Question

It is counterintuitive that a heterozygous change in a gene associated with dominant disease is treated as equal to a heterozygous change in a gene associated with recessive disease. This needs to be factored in and, as a minimum, a sub-analysis focusing on variants linked to autosomal dominant disease only, should be performed. In fact, it is tempting to adopt a methodology that only looks at this group.

Reply

In the revision, the following paragraph is added in Discussion to address the issue: It is particularly interesting that majority of the diseases caused by the mutated DDR genes are autosomal recessive. This can substantially diminish the impact of the

DDR deleterious variants in disease susceptibility in human population although the DDR deleterious variants can be highly prevalence, as reflected by the rarity of the diseases caused by autosomal recessive *LIG4* deleterious variation. However, the deleterious effects of mutated DDR genes could be maintained in the populations with consanguinity culture (Bittles & Black, 2010). It is also interesting to note that the deleterious variants in certain DDR genes causing autosomal dominant diseases can also be highly prevalent in human populations. This is represented by the high cancer-risky deleterious variants in *BRCA1* and *BRCA2* that the carrier rate is one in a few hundreds of individuals in general population, for example, one in 384 in Japanese population (Momozava et al, 2018), one in 265 in Chinese Han and Mexican populations (Dong et al, 2021, Fernández-Lopez et al, 2019), one in 256 in Malaysian population (Wen et al, 2018), one in 189 in US population (Manickam et al, 2018), and the highest of one in 46 in Ashkenazi Jewish population (Gabai-Kapara et al, 2014). Besides their deleterious effects, there must be beneficial significance for the high prevalence of high-risk genetic predisposition in human population. In contrast to the stable status in most of species, human BRCA is under strong positive selection leading to its high variability of over 70,000 variants identified so far (Huttley et al, 2000; Cline et al, 2018). Besides the classical function of DNA damage repair, *BRCA* gains multiple new function including including regulation of immunity against viral infection (Lou et al, 2014), gene expression regulation (Rosen et al, 2006), promoting neural development (Pao et al, 2014), and enhancing reproduction (Smith et al, 2013).

Question

The authors need to account for and/or discuss ascertainment bias. It seems that ClinVar was used as a key filter but of course groups from Europe and North America are more likely to deposit data in this repository.

Reply

Fully agree. In the Discussion, we clearly stated the limitation “One limitation of our study is the lack of sufficient DDR data from non-European populations. It reinforces the importance of studying diverse populations in human genetic study (Sakaue *et al*, 2021; Sirugo *et al*, 2019).”. Ideally, it would be more solid for the study to include the data from as many as possible global ethnic populations. However, the reality is that most of the data in current genomic databases are dominated by european ascent population. For example, our analysis of BRCA variation data confirmed this issue as shown by the following figure (PMID: 30702160).

While we tried hard to collect all possible genomic data for our study, we were only be able to reach the current state of including 16 ethnic populations in the study. It is certainly a bias in human genetic study. In fact, we are working hard in studying genetic predisposition in asian population (PMID: 32467295, PMID: 34235180, PMID: 35165121, PMID: 32817299; PMID: 32963034, PMID: 26848529).

Question

The article contains a number of typos and imprecise descriptions and would benefit from further proof reading. I have highlighted some of these issues that were encountered in the introduction to point to this issue but highlighting these inaccuracies throughout the manuscript is beyond the scope of this review.

Reply

In the revision, we tried hard to improve the quality of the english writing. We have made correction for each error pointed out by the reviewer as the followings. The final version was also edited by a native English speaker colleague. We hope the quality of the English is better than the original submission.

INTRODUCTION

** replace "the phenonium" with "this phenomenon" - corrected.

** "it has seldomly been tested by the actual deleterious data from human populations." (unclear/imprecise phrasing - please amend)
– revised as “Although this concept is attractive in explaining the relationship between deleterious variants and diseases, the evidence was largely indirect as they were mostly based on the deleterious variants predicted by in silico prediction tools, which is well determined as tending to over-predict the deleteriousness of genetic variants (Cubuk *et al*, 2021; Richards *et al*, 2015). For the studies used the deleterious variants identified from human origin, the results were often restricted by the limited data quantity (Fu *et al*, 2013), restricted population size (Lohmueller, 2014), or limited to the populations with specific diseases (Huang *et al*, 2018). Therefore, it remains largely unclear for the distribution patterns of deleterious variants in human populations.”

** replace "existing evidences" with "existing evidence"
– corrected as “evidence”

** replace "current arts" with "current tools"
– deleted “current arts”

** replace "questionable that they tend" with "questionable as they tend"
– replaced by “which is well determined as tending to over-predict the deleteriousness of genetic variants”

** "Certain studies indeed used the actual deleterious variants derived from human origin." (unclear/imprecise phrasing - please amend).

– revised as “For the studies used the deleterious variants identified from human origin, the results were often restricted by the limited data quantity (Fu *et al*, 2013), restricted population size (Lohmueller, 2014), or limited to the populations with specific diseases (Huang *et al*, 2018).”

** "However, many DDR genes themselves are prone to germline variation, in which a part is deleterious leading to high risk of diseases" (unclear/imprecise phrasing - please amend).

– “However, many DDR genes are prone to germline variation, a part of which can be deleterious in causing increased risk of various diseases including cancer.”

** replace "Because of their medical values" with "Because of their medical value".

– revised as “Because of their medical value”

** replace "have been studied in very details" with "have been studied in great detail".

– revised as “have been studied in great detail”

** replace "characterized at the populational level" with "characterized at the population level".

– revised as “characterized at the population level”

** "Taking the advantages, we used the deleterious variation data in human DDR genes as a model to test the concept of deleterious load in human populations." (unclear/imprecise phrasing - please amend).

– revised as “In this study, we used deleterious variants in DDR genes as a model to study deleterious variants in human populations”

** "highlighting that ethnic-specific deleterious variants rather than the loading is the major determinant for the differential disease susceptibility in human populations." (unclear/imprecise phrasing - please amend)

– revised as “Our study indicates that DDR deleterious variants is highly ethnic-specific. It may contribute to different disease susceptibility in different human populations.”

RESULTS

** "The 9 DDR pathways contain a total of 276 DDR genes composed of 169 distinct DDR genes": it is unclear whether the number of genes is 276 or 169. Please amend/clarify.

– A group of the DDR genes are involved in multiple DDR pathways. Therefore, only 169 distinct genes in the 276 DDR genes counted from the 9 DDR pathways.

** the term "deleterious" is used in the introduction while the term "pathogenic" is predominantly used in the results. The authors should use consistent terminology, provide definitions and justify their choice of terms. For example, as per MacArthur *et al* (Nature 2014, DOI: 10.1038/nature13127) deleterious is a variant that reduces the reproductive fitness of carriers (and would thus be targeted by purifying natural selection). In the same paper, the term pathogenic is used to define

a variant that contributes mechanistically to disease, but is not necessarily fully penetrant (i.e., may not be sufficient in isolation to cause disease). I propose that the authors introduce and consistently use the term "presumed pathogenic" or "presumed functional" to describe the group of variants that they considered.

– thanks for this valuable comments. Indeed, we were struggled in determining the term used to describe the variants. Based on the reference the reviewer provided, we consider it better to use “deleterious variants” as the standard term in our study. This is based on the consideration that all variant data we collected were from general population rather than disease cohort. Therefore, the data reflects more the presence or absence in general population but not about variant penetrance and disease risk. “Pathgenic” would be kind of misleading. Although ClinVar was used in the study, it is among many annotation references and mainly for clinical classification reference to locate the “pathogenic” “likely pathogenic” variants.

** In the methods, it seems that Clinvar was the primary source used for assigning functional significance to variants; it would be useful to include a mention to this in the results.

– Revised as “Clinical significance of each variant was referred to ClinVar classification as “Pathogenic”, “Likely pathogenic”, Variants of Uncertain Significance (VUS), Likely benign, and Benign by referring to ClinVar”

**Similarly, a mention to the source of the population data would enhance the readability of the manuscript.

- The source of the population data are included under Materials and Methods: “Genetic variants in DDR genes of general human populations were collected from the following resources: Chinese population from the China Metabolic Analytics Project (ChinaMAP) (Cao *et al*, 2020, <http://www.mbiobank.com/>, accessed in September 9th, 2020); Japanese population from the 3.5KJPNv2 (Tadaka *et al*, 2019; <https://www.megabank.tohoku.ac.jp/english/about-the-change-on-the-release-of-3-5kjp/>, accessed September 23rd, 2020) ; Korean population from the Korean Variant Archive (KOVA) (<http://kobic.re.kr/kova/>, accessed September 29th, 2020) and gnomADv2 non-cancer data (Karczewski *et al*, 2020; Lee *et al*, 2017; <https://gnomad.broadinstitute.org/>, accessed December 16th, 2020); Icelander population from the deCODE (<https://www.ebi.ac.uk/eva/?eva-study=PRJEB15197>) after filtered by the variant data from Icelander patients (Gudbjartsson *et al*, 2015; Jonsson *et al*, 2017; <https://www.ebi.ac.uk/eva/?eva-study=PRJEB8636>, accessed September 26th, 2020); variation data of non-Finnish European (Estonian, Bulgarian, Swedish, Southern European, North-Western European and Other Non-Finnish European), Finnish, Latino/Admixed-American, Ashkenazi Jewish, African/African-American, South Asian, other East Asian were extracted from gnomADv2 noncancer data. Only the variants marked as “PASS” in the corresponding VCF file were used in our study. The genome position of variants was based on human reference genome sequences hg38.”

** the term "somatic pathogenic variants" implies that the changes were not detected in the germline. It would be clearer if the description "presumed pathogenic, COSMIC-listed variants" was used instead. – In the revision, we consider that the

issue of somatic variants is less relevant to the focus of our study on germline variation. Therefore, we remove this part in the revision.

DISCUSSION

** "In contrast to the widely belied view, the results from our current study provide evidence to show that the load of deleterious variants is at similar level in different human populations due possibly to a tolerating threshold controlled under tight evolution selection, whereas the spectrum of deleterious variants is highly ethnic-specific in reflecting human adaptation to different local environments". (unclear/imprecise phrasing with a tendency to overinterpret the results; please amend and include references when discussing the widely accepted view).

– We fully agree with reviewer's comments that there is lack of physical evidence to link the DDR deleterious variation to disease susceptibility in human population, although the possibility could not be ruled out. Therefore, we focused only on the general patterns of DDR variation but not their impact on disease susceptibility. As such, we revised the paragraph as "Data from our study provide two important observations: 1. DDR deleterious variants were loaded at similar levels in human populations centered at 21 per 1,000 individuals. As deleterious variants can cause genome instability, they must be present at tolerable threshold under tight evolution selection pressure. Exceptions were the populations with smaller size or unique evolution history. 2. DDR deleterious variants in human populations were highly ethnic-specific. This reflects the genetic diversity of human populations from their adaptation to their natural environments.

** "Therefore, the different susceptibility of human populations to disease as represented by DDR deleterious variants can be better explained by the differences of deleterious variant spectrum than the different level of deleterious variant load." (the authors did not study the correlation between variation and prevalence of disease, thus the above conclusion is not substantiated by the study results [see also Major issues section]).

- IBID

** A mention to the age of genetic variation is included in the third paragraph of the discussion. Further analysis to look into this important topic could be performed using tools like <https://human.genome.dating/>.

– in this study, we didn't address the issue of genetic variation age. We indeed studied this interesting issue in our other study (PMID: 35165121).

** "The absence of the founder mutation BRCA2 999del5 in Icelander population highlights that it is at lower prevalence level in general population but enriched in Icelander breast cancer cohort" (this is a confusing statement; please clarify and include references; also please use HGVS nomenclature when describing genetic variants).

–revised as "*BRCA2* c.771_775del (999del5) is the major founder mutation in Icelander breast cancer (Thorlacius et al, 1997). However, it was not present in the 27 DDR deleterious variants identified in the Icelander population of 12,584 individuals included in our study. Its absence in Icelander general population

highlights the possibility that it has lower prevalence in Icelander general population but enriched in Icelander breast cancer cohort (Tulinius *et al*, 2002).”

MATERIALS and METHODS

** More information is required on how the ancestral groups were determined. There is a mention to the 'source of variation data' in the first section of the Methods but it would be worth providing more details (e.g. were all populations defined using PCA-based analysis?)

- In each of the original studies, the ancestry for each population was tested by either principal component analysis (Chinese, Japanese, Korea, gnomADv2) or genotyping (Icelander) as indicated in the original studies.

** "variation data of non-Finnish European (Estonian, Bulgarian, Swedish, Southern European, North-Western European and Other Non-Finnish European)" (please clarify the source of these data - would it be fair to presume that gnomADv2 was used?).

– Correct. variation data of non-Finnish European (Estonian, Bulgarian, Swedish, Southern European, North-Western European and Other Non-Finnish European), Finnish, Latino/Admixed-American, Ashkenazi Jewish, African/African-American, South Asian, other East Asian were extracted from gnomADv2 noncancer data.

** "Pathogenic and Likely pathogenic variants (PVs) were defined as deleterious variants and used in the study." (the abbreviation PV is used extensively in the text and should be properly define. It sounds like the authors considered all changes listed as pathogenic or likely pathogenic in Clinvar as functional and included them in a group that they used for further analysis. If that is the case, I would advice using the term "presumed pathogenic" or "presumed functional" to describe this subset of genetic variants. – As discussed above, we considered it better to use “deleterious variants” as the standard term in our study. This is based on the consideration that all variant data we collected were from general population rather than disease cohort. Therefore, the data reflects more the presence or absence in general population but not about variant penetrance and disease risk. “Pathgenic” would be kind of misleading. Although ClinVar was used in the study, it is among many annotation references and mainly for clinical classification reference to locate the “pathogenic” “likely pathogenic” variants.

** "p<0.05 was considered as statistically significant." (it is likely that a degree of multiple testing was involved; how did the authors deal with this important issue?)

- Following the comments, we performed Benjamini-Hochberg procedure for Chi-square (χ^2) and t-test results. We observed that deleterious variant-affected DDR genes were only enriched in Base Excision Repair pathway but not in others (Table 1A), whereas the deleterious variant loads remain widely differences between different populations (Table 2). Thanks very much for this valuable comment to avoid misleading information.

Reviewer #2

Question

- The conclusion that PV frequencies resemble between two kin ethnic groups is not a novel finding. In fact, to reach this conclusion, more rigorous way would be to analyze all variants including single nucleotide polymorphisms (i.e., benign variants).

Reply

This is indeed the case. This issue has been extensively analyzed by previous population genomic studies including the haplo-project. The focus of our study is the DDR pathogenic variation, which by definition is deleterious to genome stability, therefore, are under strong evolution selection pressure as indicated by the first reviewer. As shown by the data from our study, the features of PVs is substantially different from the polymorphism in normal populations, which is largely beneficial, neutral but less deleterious on genome stability.

Question

- This report lacks association of the genetic variations with clinical consequences. Is the Bulgarian population predisposed to certain types of cancers due to the high PV? Or does the Ashkenazi Jewish population have lower risks of diseases caused by the DDR genes? It is a little counter intuitive as Ashkenazi Jewish population often is represented with a unique spectrum of genetic diseases.

Reply

We fully admit that we don't have convincing data to support the association of the genetic variations with clinical consequences. The same issue was raised by the Reviewer 1. In the revision, we revised the focus of the study in the variation data but not their link to diseases: "there was no direct data in our study in linking the DDR variation to disease susceptibility. Following the comments, we performed extensive searching to find the potential supportive epidemiological data. We found that the most relevant data are the WHO cancer statistics. However, the data are based on the IC10 system, which provide the overall data from each cancer type but doesn't separate further into hereditary (germline, around 10% of total cases attributed with germline mutation) or sporadic (somatic, the majority of cancer cases attributed with somatic mutation) cancer. Our study is focused on germline variant, without the actual data from hereditary cancer, it is difficult to link the data to the DDR data in different populations. We fully agree with Reviewer's comments that it is too ambitious for the study to link genotype with phenotype in human populations without actual evidence as it overinterpretes the data to disease susceptibility, instead to put the focus of the study on the general patterns of DDR variation in human populations."

Bulgarian population predisposed to certain types of cancers due to the high PV?

– We can't find the actual data to make the conclusion so far, largely due to the population cancer data mixed hereditary cancer with sporadic cancer, that the signals from lower percentage of hereditary cancer are diminished by the vast majority of sporadic cancer. For example, germline mutation caused breast cancer only account for 5% of total breast cancer cases although breast cancer is the top cancer worldwide.

Or does the Ashkenazi Jewish population have lower risks of diseases caused by the DDR genes? It is a little counter intuitive as Ashkenazi Jewish population often is represented with a unique spectrum of genetic diseases.

– although the data from our study show that the load of DDR deleterious variants in Ashkenazi Jewish population was not at high level comparing with other ethnic population (11 in 1000 individuals, Table 1C), Ashkenazi Jewish population has its unique type of genetic defect-contributed diseases (<https://www.jewishvirtuallibrary.org/ashkenazi-jewish-genetic-diseases>). For example, the 3 founder BRCA mutations (*BRCA1* c.68_69del, *BRCA1* c.5266dup, *BRCA2* c.5946del) have high carrier frequency (2.17%) in Ashkenazi Jewish population contributing to high risk of breast and ovarian cancer (Gabai-Kapara et al, 2014).

Question

- Table 1 lists numbers of individuals participated and the numbers of PVs. This is not sufficient for a report, the report should clarify what are the major PV alleles (e.g., *BRCA1* truncation mutations) and show their allele frequencies for each ethnic group.

Reply

The information was not listed in the Table 1 by the restricted space. However, extensive information for each variant is provided in the Supplementary table 1, including type of mutation, distribution and allele frequency in different populations.

Question

- Introduction introduces DDR with "at least 9 different DDR pathways", but the following sentences describes only 7.

Reply

Not every DDR genes in the 9 DDR pathways were mutated. We identified the mutations in 169 DDR genes involved in 7 DDR pathways.

Question

- This study collected multiple sources of genetic data. How each institution analyzed the DNA sequencing should be described. For example, are they all based on the whole exome sequencing, whole genome sequencing, or based on RNAseq?

Reply

The information is included in the revision:

Whole genome sequence data were from ChinaMAP, Japanese 3.5KJPNv2, Icelander deCODE, whole exome sequence data were from Korean KOVA, and whole genome sequence, whole exome sequence, and RNA sequence data were from gnomADv2.

Question

- Page 7, the second paragraph: it discusses the possibility of an advantageous *LIG4* allele in an environment, but it is counter intuitive because it also describes *LIG4* is one of the genes shared by many ethnic groups. Are these 13 ethnic groups geographically close, or sharing a similar environment that has a health impact in any way?

Reply

There were 13 populations sharing the LIG4 c.1271_1275del, including CHN, JPN, ICE, KOR(EAS_KOR), AFR, AMR, EAS_OEA, NFE_BGR, NFE_NWE, NFE_SEU, NFE_SWE, NFE_ONF, SAS. Of these 13 populations, some are geographically closer like Chinese, Korean and Japanese in East Asia, but not all are close, such as SAS and non-Finnish european populations. Further work needs to be done in order to test if these highly shared deleterious variants could be any beneficial impact.

Question

- There are many grammatical errors including typos and misuse of words. The preposition "between" has been misused at too many places (where "among" is more appropriate). It is strongly recommended to have a professional English editor go through the entire manuscript.

Reply

In the revision, we tried hard to improve the quality of the english writing. We have made correction for each error pointed out by the reviewers. The final version was also edited by a native English speaker colleague. We hope the quality of the English is better than the original submission.

RE: Life Science Alliance Manuscript #LSA-2021-01319-TR-A

Prof. San Ming Wang
University of Macau
Faculty of Health Sciences
University of Macau
Taipa
Macau SAR, Taipa 999078
Macao

Dear Dr. Wang,

Thank you for submitting your revised manuscript entitled "Distinct landscapes of deleterious variants in DNA damage repair system in ethnic human populations". We would be happy to publish your paper in Life Science Alliance pending final revisions necessary to meet our formatting guidelines.

- please address the remaining Reviewer 2 points
- please add the Twitter handle of your host institute/organization as well as your own or/and one of the authors in our system
- please add a legend for the tables to the main manuscript

A. FINAL FILES:

B. MANUSCRIPT ORGANIZATION AND FORMATTING:

Sincerely,

Reviewer #2 (Comments to the Authors (Required)):

page 3 line 3 of the second paragraph: What is 'Life organisms'?
Introduction: 9 DDR while describing only 7 - the authors still have to explain why only 7 are listed.

Result section, the second paragraph: it explains FA pathway contains 49 DDR genes etc. It may be better to clarify what database (e.g. KEGG) these values are based on.

The followings are the revision in response to your questions:

-please address the remaining Reviewer 2 points:

Question 1: page 3 line 3 of the second paragraph: What is 'Life organisms'?

Reply: The sentence has been revised as "Organisms are equipped with a DNA damage repair (DDR) system to repair the damaged DNA."

Question 2: Introduction: 9 DDR while describing only 7 - the authors still have to explain why only 7 are listed.

Reply: Deleterious variants were identified in the DDR genes in 8 DDR pathways (not 7 by typo error), but none in the 3 genes in the Direct Reversal pathway. The sentence has been revised as the following:

From these variants, we identified 1,781 deleterious variants in 81 DDR genes (47.9% of 169 DDR genes) in 8 DDR pathways but none existed in the 3 genes in the Direct Reversal pathway.

Question 3: Result section, the second paragraph: it explains FA pathway contains 49 DDR genes etc. It may be better to clarify what database (e.g. KEGG) these values are based on.

Reply: The sentence has been revised as:

The 9 DDR pathways contain a total of 169 distinct DDR genes, in which Fanconi Anemia (FA) pathway has the highest of 49 DDR genes and directed reversal repair pathway the lowest of 3 DDR genes based on KEGG and Human DNA Repair Genes databases.

-please add the Twitter handle of your host institute/organization as well as your own or/and one of the authors in our system:

Twitter handle of my host institution, University of Macau, @umacau_

Twitter handle of the first author of the study, Zixin Qin, @qinzzixin

-please add a legend for the tables to the main manuscript:

The following Table legends have been included in the main manuscript after Figure legends:

Table 1. Summary of DDR deleterious variants

Table 2. Comparison of DDR deleterious variants between 16 ethnic populations

Table 3. Top 10 highly shared DDR deleterious variants in human populations

RE: Life Science Alliance Manuscript #LSA-2021-01319-TRRR

Prof. San Ming Wang
University of Macau
Faculty of Health Sciences
University of Macau
Taipa
Macau SAR, Taipa 999078
Macao

Dear Dr. Wang,

Thank you for submitting your Research Article entitled "Distinct landscapes of deleterious variants in DNA damage repair system in ethnic human populations". It is a pleasure to let you know that your manuscript is now accepted for publication in Life Science Alliance. Congratulations on this interesting work.

DISTRIBUTION OF MATERIALS:

Again, congratulations on a very nice paper. I hope you found the review process to be constructive and are pleased with how the manuscript was handled editorially. We look forward to future exciting submissions from your lab.

Sincerely,
